# Theoretical Insight into the Interaction between Chloramphenicol and Functional Monomer (Methacrylic Acid) in Molecularly Imprinted Polymers

**DOI:** 10.3390/ijms21114139

**Published:** 2020-06-10

**Authors:** Lei Xie, Nan Xiao, Lu Li, Xinan Xie, Yan Li

**Affiliations:** College of Food Science, South China Agricultural University, No. 483, Wushan Street, Tianhe District, Guangzhou 510642, China; xielei@stu.scau.edu.cn (L.X.); xiaonan@scau.edu.cn (N.X.); xinanxie@scau.edu.cn (X.X.); yanli@scau.edu.cn (Y.L.)

**Keywords:** density functional theory, chloramphenicol, methacrylic acid, molecularly imprinted polymers, intermolecular interaction

## Abstract

Molecular imprinting technology is a promising method for detecting chloramphenicol (CAP), a broad-spectrum antibiotic with potential toxicity to humans, in animal-derived foods. This work aimed to investigate the interactions between the CAP as a template and functional monomers required for synthesizing efficient molecularly imprinted polymers for recognition and isolation of CAP based on density functional theory. The most suitable monomer, methacrylic acid (MAA), was determined based on interaction energies and Gibbs free energy changes. Further, the reaction sites of CAP and MAA was predicted through the frontier molecular orbitals and molecular electrostatic potentials. Atoms in molecules topology analysis and non-covalent interactions reduced density gradient were applied to investigate different types of non-covalent and inter-atomic interactions. The simulation results showed that CAP was the main electron donor, while MAA was the main electron acceptor. Moreover, the CAP–MAA complex simultaneously involved N-H···O and C=O···H double hydrogen bonds, where the strength of the latter was greater than that of the former. The existence of hydrogen bonds was also confirmed by theoretical and experimental hydrogen nuclear magnetic resonance and Fourier transform infrared spectroscopic analyses. This research can act as an important reference for intermolecular interactions and provide strong theoretical guidance regarding CAP in the synthesis of molecularly imprinted polymers.

## 1. Introduction

Chloramphenicol (CAP), a broad-spectrum antibiotic that inhibits both pathogenic gram-positive and gram-negative bacteria, is widely used to treat bacterial diseases in humans and animals owing to its high efficiency, low price, and stability. However, CAP residues retained in domestic animals can enter the human body through the food chain and show potential carcinogenicity and genetic toxicity to consumers. Therefore, the sensitive detection with CAP residues in food of animal origin has attracted extensive research attention in related fields [1,2,3]. Currently, the internationally recognized methods for detecting CAP residues are gas chromatography, liquid chromatography, and chromatography-mass spectrometry [4,5]. However, these methods have considerable shortcomings, such as complicated sample pre-processing and slow analysis speed, making it necessary to develop more robust analytical methods for CAP detection with a high precision and selectivity.

Molecular imprinting is an extremely efficient class of molecular recognition technologies that can be used to create molecularly imprinted polymers (MIPs) with tailor-made binding sites capable of recognizing and rebinding the template molecules from a mixture of closely related compounds [6,7,8]. In recent years, the preparation of molecularly imprinted materials has been reported mostly with CAP as a template to detect CAP in animal-derived food samples [9,10]. The adoption of computational methods in MIPs design has permitted the screening of imprinting conditions and efficient preparation of high-affinity polymers. Liu et al. screened the template-monomer molar ratio and determined the best cross-linking agent through computer simulations to model CAP MIPs toward enhancing their adsorption and selectivity [11]. In our previous research, we also used molecular dynamics to study the effect of solvent environment on the pre-assembly of CAP MIPs toward selecting a more suitable solvent as a porogen [12].

Studies have shown that the recognition and physical properties of MIPs are dependent on the success of template‒monomer interactions during the pre-polymerization stage [13,14,15]. With the rapid development of in silico simulation, a current popular topic is the employment of computational chemistry to assist in studies on the number and relative positions of interaction sites between template molecules and functional monomers [16], which may influence the imprinted efficiency of MIPs. For instance, based on density functional theory (DFT), Zhang et al. used atoms in molecules (AIM) topology analysis to investigate the nature of template–monomer complex bonding and successfully explained the nature of the reaction occurring in the imprinting process [17]. Pei et al. analyzed the natural bond orbitals (NBOs), AIM, and reduced density gradient (RDG) of the epinephrine–uracil complex and showed that the intermolecular hydrogen bond interactions play an important role in the stabilization of the complex [18]. However, there remains a need for a systematic study on the interaction types and binding sites of the template and monomers in CAP MIPs.

In the present study, we selected the most suitable monomer from acrylic acid (AA), methacrylic acid (MAA), acrylamide (AM), and methyl methacrylate (MMA) through DFT calculations, which can non-covalently bind to the CAP template during the molecular imprinting process. The frontier molecular orbitals (FMOs), molecular electrostatic potentials (MEPs), AIM topology analysis, and non-covalent interactions RDG (NCI-RDG) were applied to predict and analyze the molecular imprinting systems. We calculated and measured the fourier transform infrared spectroscopic (FTIR) and hydrogen nuclear magnetic resonance (^1^H-NMR) by theoretical and experimental methods to validate the simulation results. Our work provides a more detailed view of the intermolecular interactions that occur between CAP molecules and their functional counterparts and offers a valuable reference for analyzing the recognition mechanism of other polymeric systems.

## 2. Results and Discussion

### 2.1. Theoretical Screening of Functional Monomers

The DFT method at the B3LYP level with the 6-31G+(d,p) basis set has been widely applied to obtain the most stable configurations and binding energies of various structures [19]. In this work, CAP was chosen as a template, and MMA, AM, AA, and MAA were chosen as functional monomers. The interaction energies (ΔE) and Gibbs free energy changes (ΔG) of the monomer–template complexes (molar ratio 1:1) in the gas phase are shown in Figure 1. The values of ΔE for the binding of CAP to the different monomers decreases in the order of CAP–MMA > CAP–AM > CAP–AA > CAP–MAA. The lower (more negative) the interaction energy is, the stronger the interaction between the template and monomer, the more stable the complex, and the greater the specificity of the MIPs binding toward the target molecule [20]. Thus, MAA is potential to be the most suitable functional monomer for targeting the CAP molecule from the screened options.

It is also worth noting ΔG of the template–monomer pre-polymerization complexes, which represents the reaction probability [21]. Though the imprinting efficiency strictly depends on ΔG in the re-binding process, consideration of ΔG in the pre-polymerization step can also provide valuable information regarding the MIPs characteristics. The smaller ΔG is, the fewer cavities there will be in the complex after polymerization, which results in a weaker molecular imprinting recognition ability of MIPs [22]. As shown in Figure 1, ΔG of MAA was higher in magnitude, indicating that the process more readily occurs spontaneously. Therefore, MAA was selected as the suitable functional monomer for subsequent experiments.

### 2.2. The Effect of Solvent

In the non-covalent imprinting systems, the solvent environment has an important influence on the formation of stable complexes between template molecules and functional monomers. The selected solvent should not only have a high solubility to the components, but also have less interference with the non-covalent interaction between template and monomer [17]. In this section, three solvents, including methanol, acetonitrile and chloroform, were selected to analyze the impact on the CAP and MAA.

As shown in Figure 2, the order of the solvation energy of CAP and MAA in different solvents is as follow: methanol > acetonitrile > chloroform, indicating that CAP or MAA has stronger interaction with methanol than with acetonitrile and chloroform. However, in the molecularly imprinted polymer synthesis process, when a solvent having a high solvation value was selected as a porogen, the strong interaction between the template molecule and the solvent shields the molecular interaction sites and weakens the interaction with the functional monomer, making the molecular recognition ability of the imprinted polymer relatively poor. According to the calculation results in Figure 2, CAP and MAA have small solvation energy values in the chloroform environment, indicating that chloroform has a weak destructive force on template-monomer interactions. Therefore, chloroform as a porogen is more suitable for the synthesis of CAP MIPs.

### 2.3. Analysis of FMOs

In frontier orbital theory, based on quantum chemistry, the interactions of molecular orbitals are affected when the molecule is involved in a reaction, and the FMOs preferentially react [23]. The FMOs include the highest occupied molecular orbital (HOMO) and lowest unoccupied molecular orbital (LUMO) play a crucial role in the reaction mechanisms. The HOMO energy (E_HOMO_) is proportional to the electron-donating ability of the molecule, i.e., its electron-donating ability increases with increasing E_HOMO_; and the LUMO energy (E_LUMO_) is inversely proportional to the electron-accepting ability of the molecule, i.e., the electron-accepting ability increases with decreasing E_LUMO_. The FMOs were analyzed after optimizing the structures of CAP and MAA by the B3LYP/6-31G+(d,p) method. Moreover, as most of the chemical reactions may be characterized in terms of electrophilic/nucleophilic action of charge transfer through accepting/donating electrons, thus the calculation of electrophilicity index (ω) becomes particularly important in the DFT [24]. The results of analysis are shown in Figure 3 and Table 1.

As shown in Figure 3, in the template molecule CAP, the electron cloud of the LUMO is mainly distributed across the entire molecule and covers the –C=O and –NH– functional groups, while in the HOMO orbital of CAP, no such arrangement was observed. This result showed that CAP may serve as an electron donor. Additionally, the –COOH– of MAA is located within both the HOMO and LUMO, indicating that MAA may act as both an electron donor and electron acceptor. Furthermore, as shown in Table 1, E_HOMO_ of CAP (−2.238 eV) is less negative than that of MAA (−2.335 eV), meaning that CAP is a stronger electron donor and is more reactive than MAA. The E_LUMO_ value of MAA is 0.824 eV, which is lower than that of CAP, confirming MAA as the main electron acceptor. The ω as a measure of the ability of a molecule to attract an electron [25], calculated for MAA (0.189) was higher than that of CAP (0.038), which indicated that the electron-accepting ability of the former is stronger than that of the latter.

### 2.4. Analysis of MEPs

In order to quickly identify the different reaction sites on the surfaces of CAP and MAA, we further analyzed their MEPs. MEP maps use the local electron charge density to systematically investigate the ability of a molecule to interact. By referring to the mapped colors, we judged the positions of strong electropositive and electronegative atoms that help to determine the active sites and determined the possible coordination modes of the template compound with the functional monomer. The MEP maps of CAP and MAA are shown in Figure 4 with different colors. Yellow represents electron-rich regions, where the electrostatic potential is negative, and blue represents electron-deficient regions, where the electrostatic potential is positive [26,27].

As shown in Figure 4, the negative region of CAP is located mainly around O12, O13, and O28, which is easily attributed to their lone pairs of electrons [28], indicating that O12, O13, and O28 of CAP have higher energy levels. Hence, they are susceptible to attacks by electrophiles and can lose electrons, making them electron donors. On the other hand, the positive region of CAP is mainly distributed around H24 and H26, revealing that the two atoms are at a lower energy level and can easily be attacked by nucleophiles, making them electron acceptors. Similarly, in MAA, the H14 atom is the electron acceptor, and O12 is the electron donor. Therefore, we can conclude that the potential active sites are O12, O13, O28, H24, and H26 for CAP and H14 and O12 for MAA, which are readily involved in both electrophilic and nucleophilic substitution reactions.

### 2.5. Topological Analysis

The topological AIM analysis calculations are directly related to any chemical bond, including hydrogen bonding, which is characterized by the existence of a bond critical point (BCP) [29]. In order to understand the nature and strength of the hydrogen bonding interactions in the CAP–MAA complex, we further investigated them with the help of the Multiwfn software based on Bader’s AIM theory [30]. The AIM graph displaying all the BCPs is presented in Figure 5. There are three BCPs (BCP79, BCP74, and BCP82) between CAP and MAA in the reaction paths of O(28)···H(42), H(21)···O(40), and H(8)···O(40), respectively. The nature of the chemical bonds and molecular reactivity is described by the total electron density ρ(r), Laplacian electron density ▽^2^ρ(r), and electron energy density H(r), which is composed of the electron potential energy density V(r). The values of the AIM topological parameters of O(28)···H(42), H(21)···O(40), and H(8)···O(40) for the complex are listed in Table 2.

Rozas et al. [31] and Fuster et al. [32] suggested that these criteria can be used to characterize the nature of interactions, i.e., when▽^2^ρ(r) > 0 and H(r) > 0 at the BCP, the molecules interact weakly (electrostatic interaction); when▽^2^ρ(r) > 0 and H(r) < 0, the molecules interact moderately (hydrogen bonding interaction); and when▽^2^ρ(r) < 0 and H(r) < 0, the molecules interact strongly (covalent interactions). As shown in Table 2, for the CAP–MAA complex, the values of ρ(r) and▽^2^ρ(r) for the three BCPs vary from 0.007 to 0.050 a.u. and 0.028 to 0.134 a.u., respectively. The▽^2^ρ(r) and H(r) values at BCP82 are positive, which showed that there is an electrostatic interaction between H(8) and O(40). The▽^2^ρ(r) and H(r) values of H(21)···O(40) and O(28)···H(42) are 0.098 and −0.00005 a.u. and 0.134 and −0.00094 a.u., respectively, revealing that the interactions at BCP74 and BCP79 are hydrogen bonding. According to Koch et al. [33] and Lipkowski et al. [34], hydrogen bonding interactions should have ρ(r) and▽^2^ρ(r) values within the ranges of 0.002–0.035 a.u. and 0.024–0.139 a.u., respectively. In contrast, in this work, the ρ(r) of BCP at O(28)···H(42) is over 0.035 a.u., indicating that the hydrogen bond is stronger than that of BCP74. Therefore, these calculated results indicate simultaneous double hydrogen bonding provided by the template and monomer in the CAP–MAA complex, which can facilitate the tight association of the auxiliary and substrate and thus appears to be a particularly effective method for polymer synthesis.

### 2.6. NCI-RDG Analysis

The NCI between the different entities of the compound were determined graphically based on the RDG. NCI-RDG analysis can provide details regarding weak NCI in real space and visualize different regions where NCI emerge. Thus, the method is capable of distinguishing hydrogen bonding, van der Waals, and repulsive steric interactions through simple color codes [35,36]. The NCI scatter diagrams and RDG isosurfaces for the CAP–MAA complex are illustrated in Figure 6. The observed red regions indicate strong repulsion, blue regions indicate strong attraction, and green regions indicate electrostatic interactions.

sign(λ_2_)ρ is the result of the sign of λ_2_ (the second largest eigenvalue of the Hessian matrix of electron density) and ρ. The RDG function combined with sign(λ_2_)ρ was used to distinguish hydrogen bond interactions from other interactions. As shown in Figure 6a, the values of sign(λ_2_)ρ range from −0.05 to 0.05 a.u. The spikes on the left correspond to negative sign(λ_2_)ρ, where the most negative value in blue is −0.029 a.u., and the spikes on the right correspond to positive sign(λ_2_)ρ, where the least positive value in red is 0.016 a.u. This suggests a combination of strong hydrogen bonding, van der Waals, and steric interactions contributing to the high stability of the CAP–MAA complex [18].

In the RDG isosurface map (Figure 6b), the green and red regions indicate the existence of van der Waals and steric effects, respectively, within the CAP–MAA complex. In addition, the blue circular intermolecular hydrogen bonding regions suggest that the CAP–MAA complex has a hydrogen bonding interaction in this region, where a bluer isosurface indicates a stronger hydrogen bond. Therefore, as shown in Figure 6, the hydrogen bond interaction of O(28)···H(42) is stronger than that of H(21)···O(40), which is more conducive to stabilizing the system. This result is consistent with the AIM analysis discussed in Section 2.5.

### 2.7. FTIR Analysis

FTIR vibrational spectroscopy is a highly sensitive technique that can identify the physicochemical interactions occurring between molecules. To verify our results and understand the nature of the interactions between the template and monomer, the theoretical and experimental IR spectra were analyzed and are presented in Figure 7. Although the theoretically calculated peaks were slightly different from the experimental data, they were all within the infrared characteristic absorption peak ranges of their functional groups and showed similar trends. These results were also reported by Saloni et al. [37]. The observed differences are due to the nature of the simulation calculation, where the CAP, MAA, and CAP–MAA models were treated as isolated systems; this automatically excluded any noise or interference that typically characterizes the experimental spectra.

In the MAA theoretical spectrum (Figure 7a), the peaks at 3354 and 1669 cm^−1^ can be attributed to the stretching vibrations of O–H and C=O, respectively. The characteristic peaks of O–H and C=O at 3481 and 1665 cm^−1^, respectively, were present in the spectrum of CAP, and the stretching vibration of N–H in CAP was also observed at 3705 cm^−1^ (Figure 7b). Furthermore, based on the theoretical spectrum of the CAP–MAA complex (Figure 7c), the N–H and C=O stretching peaks shifted slightly to shorter wavenumbers (3705 to 3684 cm^−1^ and 1669 to 1657 cm^−1^) compared with pure CAP and pure MAA, respectively. Similarly, slight changes were observed in the CAP–MAA complex spectrum with regards to the O–H and C=O peaks at 3324 and 1657 cm^−1^, respectively, compared with the purity monomer and template. This is strong evidence for the formation of hydrogen bonds that affect the distribution of electron clouds on the molecular functional groups, which, in turn, results in a shift in the characteristic peak to a lower wavenumber [38].

As in the theoretical spectra, the experimental data showed that the N–H, O–H, and C=O peaks all shifted to lower wavenumbers in the complex spectrum due to hydrogen bond formation between the monomer and template. The experimental and calculated spectra showed significant resemblance.

### 2.8. ^1^H-NMR Analysis

^1^H-NMR spectroscopy is one of the most vital techniques for the structural analysis of organic compounds as it gives in-depth information about the studied complexes [39]. The chemical shifts obtained experimentally were compared to the shifts calculated theoretically using the molecular structures optimized at the DFT/B3LYP/6-311G+(d,p) level of theory. The experimental (methanol-D4 solutions) and theoretical ^1^H-NMR spectra of MAA, CAP, and CAP–MAA are shown in Figure 8.

As shown in Figure 8, in the CAP‒MAA complex spectrum, the peak shift associated with the carboxyl hydrogen atom of MAA changed significantly compared with purity MAA, moving downfield from 5.71 to 5.86 ppm. The peak related to the amino hydrogen atom of CAP also shifted slightly upfield from 5.67 to 5.65 ppm. Similarly, in the experimental spectrum, the chemical shifts of the functional groups before and after the template and monomer reaction showed the same trends. A possible explanation for this result is that CAP and MAA combine by hydrogen bonding to form a stable complex, which affects the electron cloud density distribution around the amino hydrogen atom of CAP and the carboxylic hydrogen atom of MAA [40].

## 3. Materials and Methods

### 3.1. Materials

CAP (>98% purify) samples were purchased from Shanghai Aladdin Reagent (Shanghai, China), and MAA was purchased from Tianjin Damao Chemical Reagent Factory (Tianjin, China). Methanol was purchased from Guangdong Guanghua Chemical (Shantou, China), and methanol-D4 was purchased from Merck Chemical Technology (Shanghai, China).

### 3.2. Computational Details

Four widely used functional monomers, namely AA, MAA, AM, and MMA, were selected for screening. The structures of the template, functional monomers, and monomer–template complex were computationally modeled using Gaussian simulation software. Then, optimizing the structures through DFT at the B3LYP/6-31G+(d,p) basis set level [41]. The most stable configuration were obtained when the calculated molecular lattice point data in the command window is not changing and the completed state is displayed, which can be used as the initial configuration for subsequent studies.

Basis set superposition error [42,43] was included in the total interaction energy calculations [44]. Depending on the binding energy ∆E, the suitable functional monomer that binds to the template molecule was selected. Furthermore, the Gibbs free energy of the molecules was obtained through the Gaussian 09 program to judge the extent of the reaction. The interaction energies (ΔE) and Gibbs free energy changes (ΔG) were calculated by Equations (1) and (2):(1)$$\Delta E=E_{complex}-E_{template}-nE_{monomer}$$
(2)$$\Delta G=G_{complex}-G_{template}-nG_{monomer}$$

Using Tomasi Polarized Continuum Model (PCM) based on self-consistent reaction field (SCF), the solvation energies of template molecule CAP and functional monomer MAA in three common solvents were calculated. The method model describes the solvent as a uniform continuous medium. The interaction between solvent and solute was calculated by using a solute interface, which is closer to the real shape of solute molecule. The solvation energy values of CAP and MAA in three different solvents were calculated using Equation (3):(3)$$E_{in\;solvation}=E_{in\;gas}-E_{in\;solvent}$$
where E_in solvation_ is the solvation energy of the system, E_in gas_ is the single point energy in a gas phase environment, and E_in solvent_ is the single point energy for solvent conditions.

Based on the DFT method, FMO and MEP analyses were performed to predict the active sites of the template CAP and suitable monomers with their optimized structures. The electrophilicity index (ω) was calculated according to the energy of the FMOs according to the Equations (4)–(6):(4)$$\chi =-\mu =\frac{\left(I+A\right)}{2}$$
(5)$$\eta =\frac{\left(I-A\right)}{2}$$
(6)$$\omega =\frac{\mu^{2}}{2\eta }$$
where I is the ionization potential, which is approximately equal to −E_HOMO_; A is the electron affinity, which is approximately equal to −E_LUMO_; χ is the Mulliken electronegativity; μ is the electron chemical potential; and η is the absolute hardness.

For a better understanding of the interactions in the template–monomer complex, Multiwfn [45], a powerful program for wave function analysis, was used to plot the AIM and NCI-RDG for analyzing the optimized geometries. By combining these two methods, we can clearly distinguish the van der Waals, hydrogen bonding, and steric repulsion interactions in the complex. The RDG function is defined as Equation (7):(7)$$RDG=\frac{1}{2{\left(3\pi^{2}\right)}^{1/3}}\frac{|\nabla \rho \left(r\right)}{\rho {\left(r\right)}^{4/3}}$$
where |▽ρ(r)| is the mode of the electron density gradient and ρ(r) is the electron density.

### 3.3. FTIR Analysis

Theoretical and experimental FTIR analyses were performed to explore the intermolecular interactions between the template molecule CAP and the functional monomer MAA. The detailed procedures were as follows.

Theoretical calculation method: The molecular structure models of CAP, MAA, and the CAP–MAA complex were calculated by the GaussView 5.0.9 program (Gaussian, Inc.,Wallingford, CT, USA) with the B3LYP/6-31G+(d,p) basis set. All theoretical IR spectra were plotted from the Multiwfn program.

Experimental method: The CAP–MAA complex with a molar ratio of 1:4 in methanol was prepared by stirring in a 65 °C water bath for 6 h, followed by drying in an oven at 65 °C for 6 h. Next, 0.001 g of the CAP‒MAA complex, 0.001 g of CAP, and 0.001 g of MAA were used to prepare KBr pellet samples for FTIR spectral measurements (Vertex 70, Beijing, China). All the FTIR measurements were carried out at room temperature and measured from 500 to 4000 cm^−1^.

### 3.4. ^1^H-NMR Analysis

The interactions between the template and monomer molecules were also investigated using theoretical and experimental ^1^H-NMR in this work. The results were acquired by the following methods.

Theoretical calculation method: All calculations were performed with the Gaussian 16 software (Gaussian, Inc.,Wallingford, CT, USA). The geometries of the molecules were optimized at the B3LYP/6-31G+(d,p) level, and dispersion correction was used to obtain reasonable conformations. Referring to Pierens [46], the scaling method was used to obtain the NMR shifts. Based on the optimized structures, the NMR shift was calculated at the B3LYP/6-31G(d) level using the gauge-independent atomic orbital method. The PCM solvent model of methanol was used during the calculation.

Experimental method: methanol-D4 was used as the solvent to prepare approximately 5 mmol L^−1^ solutions of MAA, CAP, and the CAP–MAA complex, and the solutions were subsequently sonicated for 5 min. After standing overnight, the solutions were analyzed using a Bruker AMX 600 MHz spectrometer (Bruker, Karlsruhe, Germany) at room temperature to obtain the ^1^H-NMR spectra.

## 4. Conclusions

In this work, MAA was confirmed as the most appropriate functional monomer for CAP MIPs among AM, MAA, AA, and MMA based on ΔE and ΔG. The FMO and MEP analysis results for the template molecule CAP and functional monomer MAA showed that CAP was the main electron donor with the active sites of O12, O13, O28, H24, and H26, while MAA was the main electron acceptor with the active sites of H14 and O12. Moreover, in order to investigate the types of intermolecular interactions between the CAP molecule and MAA molecule, AIM and NCI-RDG were analyzed after obtaining the most stable configuration of the CAP–MAA complex. The simulation results showed that the complex was characterized by strong hydrogen bonding, van der Waals, and steric interactions, all of which contribute to the high stability of the imprinting system. In addition, we found that O(28)···H(42) and H(21)···O(40) of the CAP–MAA complex can simultaneously provide double hydrogen bonds, where the hydrogen bond of the former is stronger than that of the latter. The ^1^H-NMR and FTIR theoretical and experimental analyses also indicated that the interaction between CAP and MAA was hydrogen bonding. This study provides a more detailed view of the intermolecular interactions occurring between CAP molecules and their functional counterparts and provides theoretical guidance for designing more precise recognizable sites and imprinting sites to increase the specific binding capacity of MIPs.

## Figures and Tables

**Figure 1 ijms-21-04139-f001:**
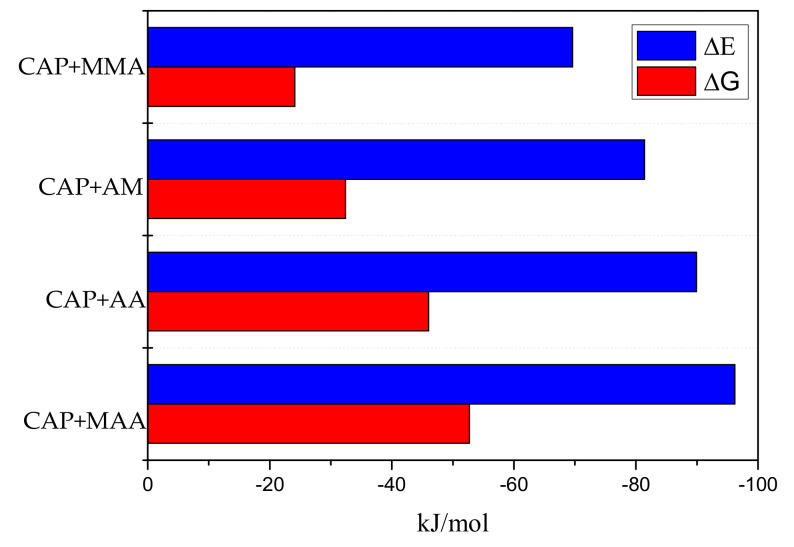
Interaction energies (ΔE) and Gibbs free energy changes (ΔG) of chloramphenicol+methyl methacrylate (CAP+MMA), chloramphenicol+acrylamide (CAP+AM), chloramphenicol+acrylic acid (CAP+AA), chloramphenicol+methacrylic acid (CAP+MAA) complexes.

**Figure 2 ijms-21-04139-f002:**
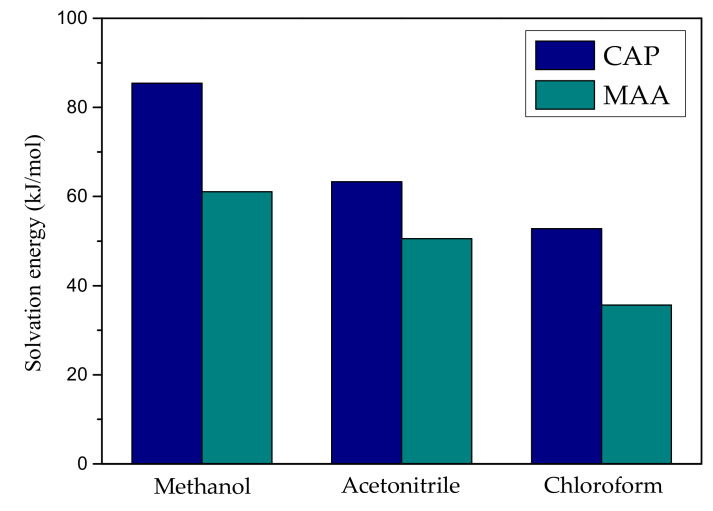
Solvation energy of CAP and MAA in different solvents.

**Figure 3 ijms-21-04139-f003:**
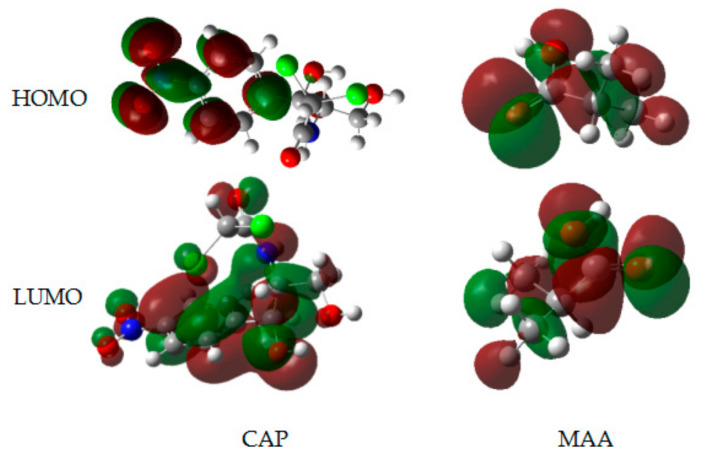
Frontier molecular orbitals (FMOs) and energy levels for highest occupied molecular orbitals (HOMOs) (top) and lowest unoccupied molecular orbital (LUMOs) (bottom) of the template CAP and monomer MAA.

**Figure 4 ijms-21-04139-f004:**
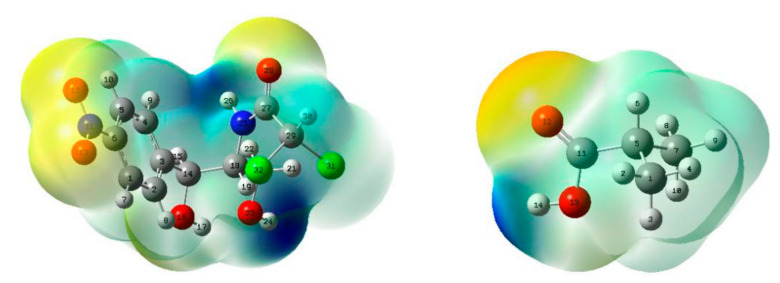
Molecular electrostatic potential (MEP) maps of CAP and MAA.

**Figure 5 ijms-21-04139-f005:**
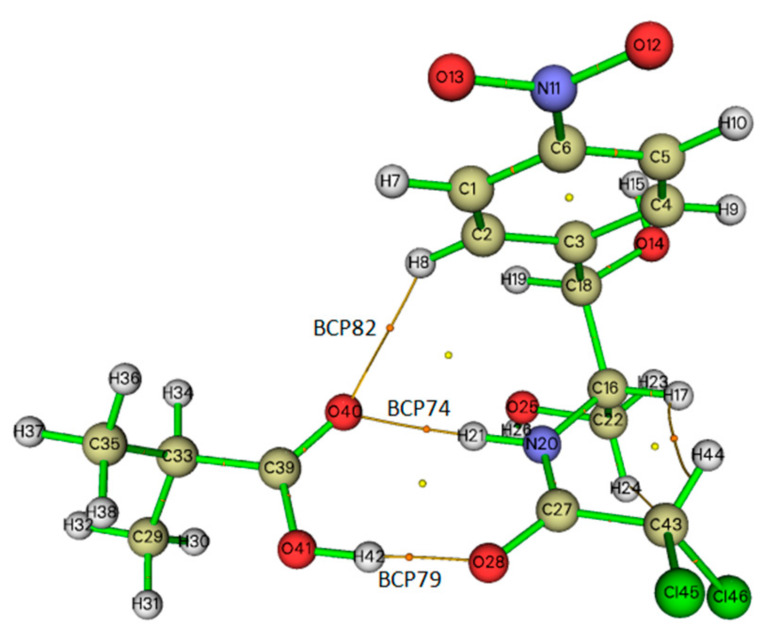
Topological atoms in molecules (AIM) graph of the CAP–MAA complex.

**Figure 6 ijms-21-04139-f006:**
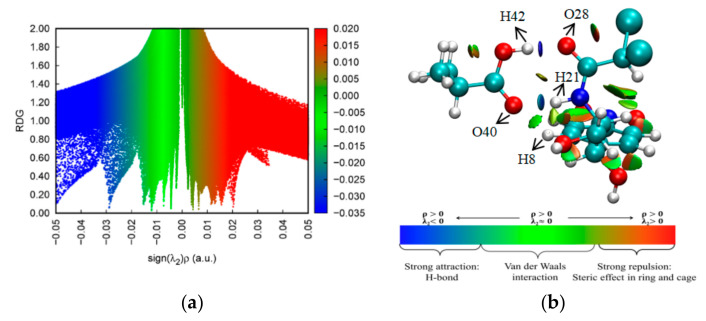
(**a**) Non-covalent interaction (NCI) scatter diagram and (**b**) reduced density gradient (RDG) analysis of the CAP–MAA complex.

**Figure 7 ijms-21-04139-f007:**
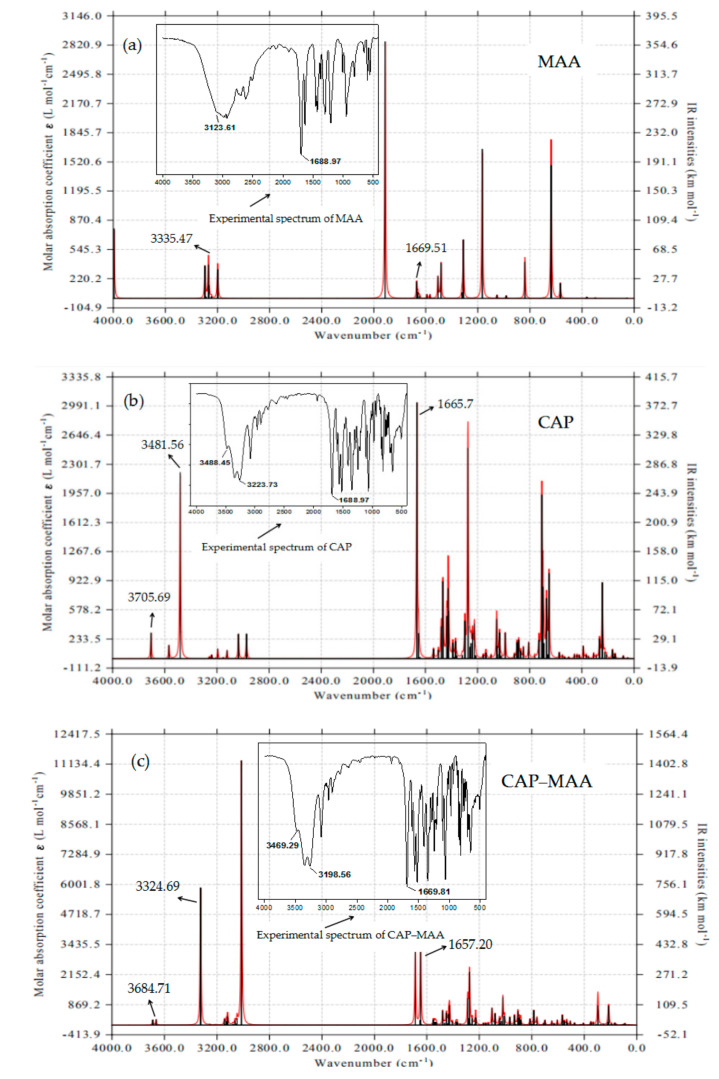
Theoretical and experimental fourier transform infrared spectroscopic (FTIR) of (**a**) functional monomer MAA, (**b**) template CAP, and (**c**) CAP–MAA complex.

**Figure 8 ijms-21-04139-f008:**
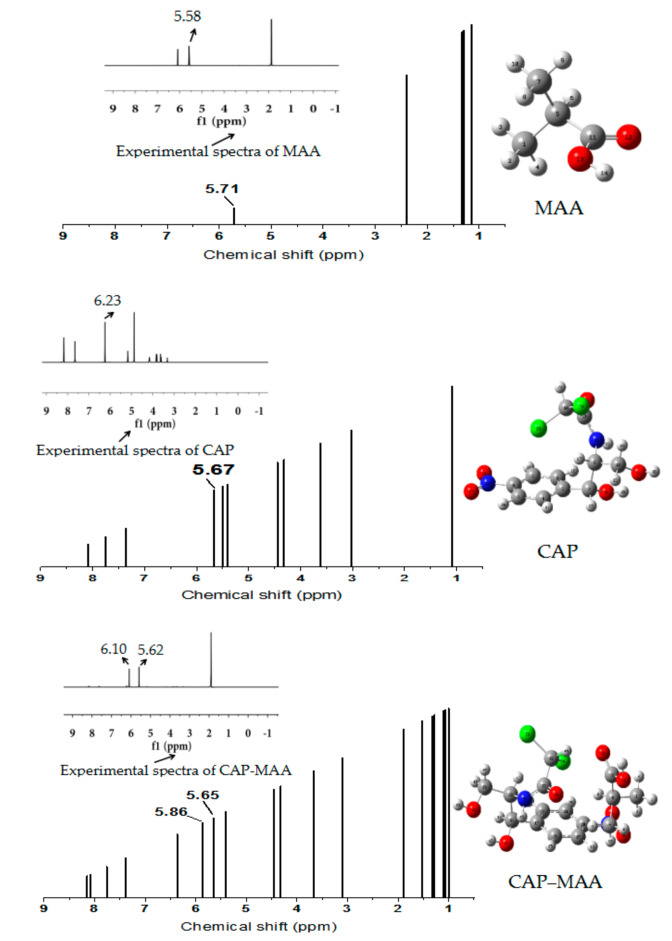
Theoretical and experimental ^1^H-NMR spectra of MAA, template CAP, and the CAP–MAA complex.

**Table 1 ijms-21-04139-t001:** Frontier orbital parameters: E_HOMO_, E_LUMO_, and ω of CAP and MAA.

Species	*E*_HOMO_ (eV)	*E*_LUMO_ (eV)	ω
CAP	−2.283	1.377	0.038
MAA	−2.335	0.824	0.189

**Table 2 ijms-21-04139-t002:** Electron and energy densities of the CAP–MAA complex at the hydrogen bond critical points (BCPs).

Complex	BCP	ρ(r) (a.u.)	▽^2^ρ(r) (a.u.)	V(r) (a.u.)	H(r) (a.u.)
CAP-MAA	BCP82, H(8)···O(40)	0.007	0.028	−0.004	0.00112
	BCP74, H(21)···O(40)	0.028	0.098	−0.024	−0.00005
	BCP79, O(28)···H(42)	0.050	0.134	−0.045	−0.00094

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
