# Peer review of "Theoretical Insight into the Interaction between Chloramphenicol and Functional Monomer (Methacrylic Acid) in Molecularly Imprinted Polymers"

_ijms, 2020, doi:10.3390/ijms21114139_

Round 1

Reviewer 1 Report

This is an interesting investigation of the bonding patterns between chloramphenicol and several monomers modeling binding sites in polymers. The paper is well organized and the results are sound and interesting. Thus, I can recommend publication.

I only have a few minor comments:

  • It is not clear how the initial configurations for the computational study have been selected. I guess that different binding configurations can be found for CAP with each of the investigated monomers, all of them being local minima. How did the authors select the final configurations to be studied? This point should be clarified in the paper.
  • It can also be useful to report a schematic representation of the structures that are considered and/or to report the coordinates of the systems.
  • All calculations have been performed in vacuum. However, the real systems are certainly considered in solution. The hydrogen bonds (and other electrostatic bonds) can be substantially influenced by a surrounding solvent. Thus, it would be appropriate to discuss the effect of a solvent on the results.

Reviewer 2 Report

11: MIPs are not method! it should be written: molecularly imprinted technology, or molecular imprinting, or development of MIPs

12: repetitive word in one sentence

14: efficient MIP for what?

I would write for example: for synthesizing efficient MIPs for recognition and isolation of CAP

20 acceptor

35: "with" instead of "of"

38: I would not write "some". "some" can be ignored. but for example " considerable" or "important" or whatever more emphasizing would be better here. because their work is based on this theory and it should be convincing enough

47: italic form for all "et al."s

62: I did not understand this sentence. does it refer to "electrostatic interactions" which are a type of "closed-shell interactions"??!

63: of the template

64: for the better English, I would suggest to rewrite this sentence: first "we selected the...", and then " from the functional monomers..."

69: "also" seems excess here

70: "we believe that" is excess, not scientific

73: polymeric systems

84: is potential to be

85: or "targeted CAP molecule"??! or "targeting the CAP molecule"?

87: it seems the "imprinting efficiency" is more harmonized with the sentence meaning

90: imprinting is not an effect. it can be written "molecular imprinting recognition (or whatever) ability of MIPs"

94: in this figure, the delta-G for CAP+MAA is less than the others. around -53. am I wrong?? or maybe the negative value is not important in this case! not sure; they wrote this complex had the highest delta-G!

99: "include" seems more transparent than "called" here

113: acceptor also 312

235: "to" instead of "with"

249: purity. right??

315: two repetitive unnecessary words "interactions"

Author Response

1. Line 11: MIPs are not method! it should be written: molecularly imprinted technology, or molecular imprinting, or development of MIPs.
Response:
Thanks very much for your helpful suggestion. We have corrected this sentence in revised manuscript, as shown below:
Molecular imprinting technology is a promising method for detecting chloramphenicol (CAP), a broad-spectrum antibiotic with potential toxicity to humans, in animal-derived foods.

2. Line 12: repetitive word in one sentence.
Response:
Thank you for reminding. We have rewritten this sentence, as shown in question 1.

3. Line 14: efficient MIP for what? I would write for example: for synthesizing efficient MIPs for recognition and isolation of CAP.
Response:
Thanks for your comments. We have corrected the expression according your example in revised manuscript, as follow:
This work aimed to investigate the interactions between the CAP as a template and functional monomers required for synthesizing efficient MIPs for recognition and isolation of CAP based on density functional theory.

4. Line 20 acceptor.
Response:
I am very sorry for my carelessness. We have corrected the spelling of this word in revised manuscript.
5. Line 35: "with" instead of "of".
Response:
Thanks for your suggestion. We have replaced “of” with “with” in revised manuscript.
6. Line 38: I would not write "some". "some" can be ignored. but for example " considerable" or "important" or whatever more emphasizing would be better here. because their work is based on this theory and it should be convincing enough.
Response:
Thanks for your helpful suggestion. The word of “some” have been changed into
“considerable” in revised manuscript according to your advice.

7. Line 47: italic form for all "et al."s
Response:
Thanks for your reminding. We have corrected the form of all “et al.”s to italic in revised manuscript.

8. Line 62: I did not understand this sentence. does it refer to "electrostatic interactions" which are a type of "closed-shell interactions"??!
Response:
Thank you for carefully and patiently reviewing our manuscript. I may not summarize the meaning of the reference accurately and now we have corrected the expression in revised manuscript. As shown below:
Pei et al. analyzed the natural bond orbitals (NBOs), AIM, and reduced density gradient (RDG) of the epinephrine-uracil complex and showed that the intermolecular hydrogen bond interactions play an important role in the stabilization of the complex.

9. Line 63: of the template.
Response:
Thanks for your reminding. We have corrected it as your suggestion in revised manuscript.

10. Line 64: for the better English, I would suggest to rewrite this sentence: first "we selected the...", and then " from the functional monomers..."
Response:
Thanks for your helpful suggestion. We have rewritten this sentence in revised manuscript, as shown below:
In the present study, we selected the most suitable monomer from acrylic acid (AA), methacrylic acid (MAA), acrylamide (AM), and methyl methacrylate (MMA) through DFT calculations, which can non-covalently bind to the CAP template during the molecular imprinting process.

11. Line 69: "also" seems excess here
Response:
Thanks for your helpful suggestion. We have deleted the “also” in revised manuscript.

12. Line 70: "we believe that" is excess, not scientific
Response:
Thanks for your helpful suggestion. We have deleted the “we believe that” in revised manuscript.

13. Line 73: polymeric systems.
Response:
Thanks for your reminding. We have corrected it in revised manuscript.

14. Line 84: is potential to be
Response:
Thanks for your reminding. We have corrected the expression of “would be” into “is potential to be” in revised manuscript according to your helpful advice.

15. Line 85: or "targeted CAP molecule"??! or "targeting the CAP molecule"?
Response:
Thanks for your comments. To make this sentence more reasonable, we have changed the “CAP target molecule” into “targeting the CAP molecule” in revised manuscript.

16. Line 87: it seems the "imprinting efficiency" is more harmonized with the sentence meaning.
Response:
Thanks for your helpful suggestion. We have replaced “imprinting effect” with “imprinting
efficiency” in revised manuscript.

17. Line 90: imprinting is not an effect. it can be written "molecular imprinting recognition (or whatever) ability of MIPs"
Response:
Thanks for your helpful suggestion. We have changed the “molecular imprinting effect” into “molecular imprinting recognition ability of MIPs” in revised manuscript.

18. Line 94: in this figure, the delta-G for CAP+MAA is less than the others. around -53. am I wrong?? or maybe the negative value is not important in this case! not sure; they wrote this complex had the highest delta-G!
Response:
Thank you for carefully and patiently reviewing our manuscript. The ΔG of CAP+MAA is -52.7 and your conclusion is correct. The spontaneity of the reaction can be evaluated by calculating the ΔG value of the template-monomer complex. The greater the absolute value of ΔG, indicating that the process more readily occurs spontaneously. In our study, the absolute value of ΔG for CAP+MAA was larger than that for others, therefore, MAA was the suitable functional monomer for CAP, which is consistent with the results of Wu et al.
Reference:
Wu, L.Q.; Sun, B.W.; Li, Y.Z.; Chang, W.B. Analyst 2003, 128, 944–949.

19. Line 99: "include" seems more transparent than "called" here.
Response:
Thanks for your helpful suggestion. The word of “called” have been changed into“include” in revised manuscript according to your advice.

20. Line 113: acceptor also 312
Response:
Thank you for carefully and patiently reviewing our manuscript. We have corrected the spelling of “acceptor” in revised manuscript.

21. Line 235: "to" instead of "with"
Response:
Thanks for your reminding. We have changed “with” to “to” in revised manuscript.

22. Line 249: purity. right??
Response:
Thanks for your reminding. The word of “pure” have been changed into“purity”in revised manuscript according to your advice.

23. Line 315: two repetitive unnecessary words "interactions".
Response:
Thank you for carefully and patiently reviewing our manuscript. We have deleted the two repetitive words "interactions".